# Comparison of Nose Wipes, Stall Sponges, and Air Samples with Nasal Secretions for the Molecular Detection of Equine Influenza Virus in Clinically and Subclinically Infected Horses

**DOI:** 10.3390/v17030449

**Published:** 2025-03-20

**Authors:** Nicola Pusterla, Kaila Lawton, Samantha Barnum, K. Gary Magdesian

**Affiliations:** Department of Medicine and Epidemiology, School of Veterinary Medicine, University of California, Davis, CA 95616, USA; kolawton@ucdavis.edu (K.L.); smmapes@ucdavis.edu (S.B.); kgmagdesian@ucdavis.edu (K.G.M.)

**Keywords:** equine influenza virus, outbreak, diagnostics, non-invasive samples, qPCR

## Abstract

In recent years, the use of non-invasive host and environmental samples for the detection and monitoring of equine respiratory pathogens has shown promise and a high overall agreement with the gold standard of nasal secretions. The present study looked at comparing nose wipes, stall sponges, and air samples with nasal swabs collected from 27 horses involved in an equine influenza (EI) outbreak. The outbreak involved 5 clinical, 6 subclinical, and 16 uninfected horses. Samples sets were collected at the onset of the index case and retested every 2–3 days thereafter until all horses tested qPCR-negative for EI virus (EIV). Nose wipes and stall sponges identified EIV in all clinical cases, and air samples identified EIV in 4/5 clinical horses. The overall agreement with all nasal swabs collected from clinical cases was 89% for nose wipes, 78% for stall sponges, and 44% for air samples. Due to the shorter shedding time in subclinical cases, nose wipes and stall sponges detected EIV in 5/6 and 4/6 subclinical horses, respectively. Only one single air sample tested qPCR-positive for EIV in a subclinical shedder. When compared to the gold standard of nasal secretions in subclinically infected horses, the overall agreement was 54% for stall sponges, 50% for air samples, and 45% for nose wipes. The collection of non-invasive contact and environmental samples is a promising alternative to nasal swabs for the detection of EIV in clinically and subclinically infected horses. However, they should always be considered as a second-choice sample type to the more accurate nasal swabs and used to test refractory horses or large populations during outbreaks. Further, the pooling of identical or different samples collected from the same horse for the qPCR testing of EIV increases the accuracy of detecting EIV, especially in subclinically infected horses.

## 1. Introduction

Equine influenza virus (EIV) is a leading respiratory virus with endemic occurrence in most horse populations [1]. In a group of susceptible horses, the morbidity rate for EI can range from 60–90% and is often lower in equids with previous exposure or vaccination [2]. EIV disrupts the respiratory epithelium in the trachea and bronchial tree, causing inflammation and subsequent development of fever, nasal discharge, and dry hacking cough [3]. However, in a population of immune horses, disease severity can be mild to subclinical, with such horses contributing to transmission and spread [4]. The diagnostic support of EI relies on the molecular detection of EIV in respiratory secretions during the acute phase of the disease [3,4]. The collection of respiratory secretions can represent a challenge when sampling refractory horses or when a large number of equids are tested for biosecurity purposes. In recent years, the use of alternative, less invasive samples, such as nose wipes, environmental swabs, and air samples, has been advocated for the testing of equine herpesviruses [5,6,7]. However, little information is presently available to support the use of alternative samples for the detection of EIV from horses and their environment. Therefore, it was the aim of this study to determine the diagnostic accuracy of nose wipes, stall sponges and air samples for the detection of EIV in infected horses and their environment.

## 2. Materials and Methods

The study population was composed of hospitalized and teaching horses stabled at a Veterinary Medical Teaching Hospital from 14 November to 19 December 2024. Patients were hospitalized for elective procedures and had no respiratory diseases prior to the outbreak. The EI suspected index case was a 9-year-old Warmblood mare presented to the hospital for evaluation of a shifting lameness. The mare developed a fever 3 days into hospitalization, bilateral nasal discharge, and coughing 4 days later. She tested qPCR-positive for EIV in nasal secretions on 14 November and was moved to isolation. The detection of this index case prompted the testing of all horses in the hospital to determine the spread of EIV. A total of 27 horses were tested for EIV at the onset of the index case and retested every 2–3 days thereafter until all horses tested qPCR-negative for EIV. Each horse was placed under infectious disease control (IDC) protocols, which included the use of disposable gloves, gowns, booties, and a footbath. EIV qPCR-positive horses with clinical signs were moved to isolation, while subclinical shedders were separated and isolated from the remaining hospitalized horses in the main barn because the number of isolation stalls was limited. No new horses were admitted into the barn until the quarantine was finished.

Nasal secretions collected via two 6-inch rayon-tipped swabs (Puritan, Guilford, ME, USA) were considered the gold standard sample type. The collection of nasal secretions followed a standard protocol for which two swabs were advanced carefully into the ventral nasal meatus and rolled for 5 s in order to collect rostral nasal secretions. The collected swabs were placed in a sterile tube and kept refrigerated with the rest of the samples until processing. Additional samples were collected and included nose wipes, environmental stall sponges and air samples. Individually wrapped 4 × 6-inch wipes soaked in saline (Professional Disposables International, Inc., Orangeburg, NY, USA) were used to wipe both nares of each horse before collecting the nose swabs. The nose wipes were then placed into 50 mL conical tubes (Fisher Scientific, Waltham, MA, USA). After collection of the nose wipes, the samples were vortexed and a small aliquot of saline solution containing the nasal secretions was collected from the bottom of each conical tube. Environmental stall samples were collected using biocide-free cellulose sponges measuring 1.5 × 3 inches and prehydrated with a neutralizing buffer diluent (3M, St. Paul, MN, USA). One sponge was used for each stall in order to collect material along the front corner where the food and automatic waterer were kept, the inside of the stall door, and the front bars of the stall that face the barn aisle for a total surface of approximately 16 square feet. Once collected, the sponges were squeezed within the bag, and a small aliquot of neutralizing buffer was retrieved for sample analysis. Air samples were collected last with a commercial Coriolis Compact air sampler (Bertin Instruments, Rockville, MD, USA). The Coriolis Compact has an airflow rate of 50 L per minute, and its dry cyclonic technology aspirates and concentrates particles and microorganisms in a disposable cone. Each air collection time lasted 8 min, keeping the instrument 12 inches from each horse and walking the respective stall so as to collect aerosolized dust. Disposable gloves were worn and changed for each sample collection. Further, the air sampler was cleaned between collections to prevent carryover contamination.

All samples were processed for nucleic acid extraction within 24 h of collection using an automated nucleic acid extraction system (Qiagen, Valencia, CA, USA) according to the manufacturer’s recommendations. Thereafter, the purified nucleic acids were tested for the presence of EIV using a previously validated qPCR assay [8]. All qPCR-positive EIV results were reported qualitatively (positive or negative). The frequency of EIV detection for the various sample types from the study horses was determined and compared. The agreement was determined as the percentage of sample pairs with identical qPCR results for EIV.

## 3. Results

Including the index case, there were 27 horses at the hospital with ages ranging from 2 to 20 years (median age 7 years). The hospital horses were composed of 14 geldings and 13 mares. Breeds included Thoroughbred (*n* = 18), Warmblood (5), Quarter Horse (2), Appaloosa (1) and pony (1). All study horses had been vaccinated against EIV within 12 months. However, information pertaining to the vaccine brands was unavailable. One of the Quarter Horses was unrelated to the hospital outbreak and presented one week after the onset of the hospital outbreak. This horse was admitted directly to the isolation barn after being shipped from out of state and developing respiratory signs.

Based on the EIV qPCR results and the presence or absence of clinical signs, the 27 horses were placed in one of three groups: clinical EI (EIV qPCR-positive horses with respiratory signs; *n* = 5), subclinical EI (EIV qPCR-positive horses without clinical signs; *n* = 6) and healthy group (EIV qPCR-negative horses without clinical signs; *n* = 16). Clinical signs observed in the 5 EI diseased horses included fever (*n* = 4; 101.7 to 103.8 °F), serous to mucoid nasal discharge (5), and spontaneous or inducible dry cough (5).

A total of 81 full sample sets were collected during the monitoring period, representing 2–7 sets per horse with a median of 2 sets per horse (Table 1). Nasal secretions detected EIV by qPCR in all clinical and subclinical cases, while all healthy horses tested qPCR-negative for EIV. The nasal secretions of clinically infected horses tested positive in 2–7 samples (median of 5 samples/horse), representing a detection period of 5–10 days (median of 9 days) from the onset of clinical disease. For nose wipes in all clinically infected horses, EIV was detected by qPCR in 2–6 samples (median of 4 samples/horse) for 2–10 days (median of 6 days). Stall sponges collected in the stalls of all clinically infected horses tested EIV qPCR-positive in 1–10 samples (median of 10 samples/horse) for 1–10 days (median of 6 days). Air samples collected in the stall of 4 EI clinical horses tested qPCR-positive in 1–2 samples (median 1 sample/horse) for 1–2 days (median of 1 day). All different sample types were able to detect EIV in at least one sample in all clinically infected horses, with the exception of air samples, which detected EIV in 4/5 horses. When compared to the gold standard of nasal secretions in clinical cases, the overall agreement of all tested samples was 89% for nose wipes, 78% for stall sponges, and 44% for air samples (Table 2). Disagreement occurred predominantly between EIV qPCR-positive nasal swabs and qPCR-negative nose wipes, stall sponges, or air samples. It took an average of 1 sample from the onset of clinical disease to detect EIV in nasal wipes and stall sponges in all clinical EI horses and 1.25 samples to detect EIV in air samples from 4/5 horses.

In subclinical EI horses, nasal secretions tested EIV qPCR-positive in 1–6 samples (median of 2 samples/horse), representing a detection period of 1–13 days (median of 2.5 days) from onset of first testing. EIV was detected in nose wipes of 5/6 subclinically infected horses in 1–5 samples (median of 3 samples/horse) for 1–9 days (median of 4 days). Stall sponges tested EIV qPCR-positive in the stalls of 4/6 subclinically infected horses in 1–4 samples (median of 2.5 samples/horse) for 1–10 days (median of 2.5 days). Only one air sample collected from the stall of a subclinically infected horse tested EIV qPCR-positive. The sample was collected on the fourth collection time and after EIV shedding had already resolved. While all nasal swabs detected EIV in subclinically infected horses, nose wipes, stall sponges, and air samples detected EIV in 5/6, 4/6, and 1/6 horses, respectively. When compared to the gold standard of nasal secretions in subclinically infected horses, the overall agreement of all collected samples was 54% for stall sponges, 50% for air samples, and 45% for nose wipes (Table 2). For this group of horses, disagreement occurred at about the same frequency between EIV qPCR-positive swabs and qPCR-negative nose wipes or stall sponges and between EIV qPCR-positive nose wipes or stall sponges and qPCR-negative nasal swabs. It took an average of 1.4 samples from the onset of testing to detect EIV in nasal wipes from 5/6 subclinically infected horses, 2.25 samples to detect EIV in stall sponges from 4/6 horses, and 4 air samples to detect EIV from 1/6 horses.

In all 16 healthy, non-infected horses, all samples tested EIV qPCR-negative for the entire monitoring period.

## 4. Discussion

The laboratory support of EI infection relies on the detection of the virus in respiratory secretions. Virus isolation via culture is known to be difficult and time-consuming and has, in recent years, been supplanted by molecular assays [3]. These so-called real-time PCR assays are highly specific and sensitive and allow for rapid testing, which is critical during an outbreak situation. Amongst various biological sample types, nasopharyngeal swabs have been shown under experimental conditions to display a higher EIV detection frequency and viral load compared to nasal swabs [9,10]. Unfortunately, nasopharyngeal swabs are often difficult to collect and poorly tolerated by equids. Therefore, the shorter 6-inch (15.24 cm) swabs are regularly collected for diagnostic purposes. The lower sensitivity of nasal swabs can be overcome by testing multiple samples from the same horse or testing multiple horses. EI outbreaks are generally seen in young naïve horses or horses with waning immunity following natural infection or vaccination. The level of immunity to EIV often determines the severity of EI disease, with no to mild clinical disease experienced by horses with preexisting immunity [1]. Preexisting immunity also impacts viral kinetics, with lower peak and shorter duration of shedding seen in horses with mild clinical disease and subclinical shedders [9]. The association between disease severity and duration of EIV shedding was observed in the present study as the 5 clinically infected horses shed virus for a median of 9 days compared to 2.5 days for subclinical shedders.

In recent studies, environmental and non-invasive contact samples have looked promising at replacing nasal or nasopharyngeal sampling. The advantage of non-invasive samples is that they are well tolerated by the patient, they do not require collection by a veterinarian, and they allow for monitoring of pathogen buildup over time. Nostril wipes, which collect dripping and dried nasal secretions around the nares, have shown 74–91% agreement with nasal swabs in the detection of equine herpesviruses in subclinical shedders [6,7]. However, agreement between nasal swabs and nose wipes depends on the duration of viral shedding and showed greater agreement for clinical versus subclinical cases. While EIV was detected by qPCR using nose wipes in all clinical cases, nose wipes detected 5 out of 6 subclinical cases. The only subclinical case that was missed using nose wipes had EIV qPCR-positive nasal secretions detected only at a single time point. Based on this observation, one would recommend the collection of more than 1 nose wipe per horse in order to detect EIV from a subclinical shedder. Multiple wipes could be collected at different time points and pooled subsequently for a single analysis. Stall sponges for the detection of respiratory pathogens have been used as a biosecurity measure to monitor the clustering of viruses at equestrian events [11,12]. The advantage of stall sponges is that they collect a large surface potentially contaminated by respiratory secretions. It is imperative to focus on areas where horses frequently interact with their environment, i.e., waterer, feed trough, and the inside of the front stall wall. However, as shown in the present study, pathogen buildup is necessary to yield EIV qPCR-positive stall sponges. This was exemplified by the higher agreement between nasal secretions and environmental sponges collected from stalls of clinical versus subclinical shedders. In the present study, only 4 out of 6 stalls of subclinical shedders tested EIV qPCR-positive in stall sponges. The two EIV qPCR-negative stalls originated from horses with the detection of EIV in nasal secretions at a single time point. Like nasal wipes, the collection of more than one stall sponge per stall of a subclinical shedder would likely increase the detection rate of EIV. EIV qPCR-positive air samples were predominantly detected in clinical EI cases during the early onset of clinical disease. This relates to the need for respiratory viruses to be aerosolized and generate high biomass in order to be detected in air samples. EIV is one of the few equine respiratory viruses that is aerosolized through the patient’s frequent coughing. In the present study, all 5 clinical EI cases displayed a dry and harsh cough during the early onset of the disease (one horse had only an inducible cough). EIV could also be detected during the dynamic air collection when walking the stall and collecting virus particles settled on the ground. This was the case in the only subclinically infected horse from which an EIV qPCR-positive air sample was collected after resolution of nasal shedding and a reminder that air samples negative for viral RNA do not necessarily mean that the virus is not present in the environment. It must be recognized that the detection of equine respiratory viruses in the air represents a challenge due to the low density of horses in the barn, the large open-air space surrounding such horses, the short shedding period, and variable virus particle size. All these factors can impact viral biomass in the air. Further, air sampling protocols lack standardization regarding airflow rate, volume of air collected, duration of sampling, and distance of the air sampler to the shedding source. The portable dry cyclonic collection device used in the present study has been successful in monitoring outdoor and indoor spaces for the presence of SARS-CoV-2 [13,14]. In the latter studies, the air collection was performed over 30 min. While more work is needed to maximize air collection protocols to detect and monitor equine respiratory pathogens, it appears that nose wipes and stall sponges are more accurate at identifying individual shedders than air sampling.

As this was a convenience study, there were various limitations pertaining to the study design, including the low number of horses involved in the EI outbreak and their immunological status against EIV. The number of samples was increased by collecting multiple sample sets from each study horse. Further, while the immune status against EIV directly impacts the peak and duration of shedding, this scenario is often encountered in populations of adult horses in North America.

## 5. Conclusions

Clinical and subclinical EIV shedders can be responsible for viral spread and introducing the virus into a population of susceptible horses. A diagnosis of EI is supported through antigen detection, ideally in nasal or nasopharyngeal samples. The testing of less-invasive contact and environmental samples has the advantage that patients experience little to no discomfort during sample collection. Further, while nasal secretions and nasal wipes detect EIV during the acute disease period, environmental samples allow for the detection of EIV beyond the shedding period, as the virus builds up and remains detectable in dried-out nasal and oral secretions. The present study showed that nose wipes and stall sponges showed a high degree of overall agreement with the gold standard of nasal swabs. The detection of EIV in air samples yielded the best results during the active shedding period of clinically infected horses. Overall, the collection of alternative host and environmental samples would be applicable to sampling refractory patients or when testing a large number of horses during an outbreak.

## Figures and Tables

**Table 1 viruses-17-00449-t001:** EIV qPCR results from 27 horses based on disease form (clinical EI (*n* = 5); subclinical EI (6) and healthy non-infected horses (*n* = 16)) and sample type (nasal secretions, nose wipes, stall sponges, and air samples).

Disease Form	Nasal Swabs qPCR pos/neg	Nose Wipes qPCR pos/neg	Stall Sponges qPCR pos/neg	Air Samples qPCR pos/neg
Clinical EI	21/6	20/7	19/8	6/21
Subclinical EI	12/10	16/6	10/12	1/21
Healthy	0/32	0/32	0/32	0/32
Total	33/48	36/45	29/52	7/74

**Table 2 viruses-17-00449-t002:** EIV qPCR agreement in paired samples from 27 horses based on disease form (clinical EI (*n* = 5); subclinical EI (6) and healthy non-infected horses (*n* = 16)) using nasal secretions as the reference sample type.

Nasal Swabs	Nose Wipes		Stall Sponges		Air Samples	
	qPCR pos	qPCR neg	qPCR pos	qPCR neg	qPCR pos	qPCR neg
qPCR positive						
Clinical EI	19	2	17	4	6	15
Subclinical EI	8	4	6	6	1	11
Healthy	0	0	0	0	0	0
qPCR negative						
Clinical EI	1	5	2	4	0	6
Subclinical EI	8	2	4	6	0	10
Healthy	0	32	0	32	0	32

## Data Availability

Data are contained within the article.

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
