# Peer review of "Comparison of Nose Wipes, Stall Sponges, and Air Samples with Nasal Secretions for the Molecular Detection of Equine Influenza Virus in Clinically and Subclinically Infected Horses"

_viruses, 2025, doi:10.3390/v17030449_

Round 1
Reviewer 1 Report
Comments and Suggestions for Authors
The authors present an interesting study where they have investigated the use of various environmental samples as potential replacements for nasal swabs for the detection of equine influenza virus (EIV).
The introductory text provides sufficient information to understand the study.
The methods are described in sufficient detail to enable replication of the study, though I have asked the authors to clarify some aspects of the methodology.
The results are well described and support the conclusions drawn by the authors.
Line 16 suggest revision “EI virus (EIV)”
Line 55 suggest adding the day the suspected index case first tested positive by qPCR.
Line 62 I would suggest the authors add clear definitions for each of the three groups of horses. My understanding is “cases” were defined as those horses exhibiting clinical signs of respiratory disease and testing positive for EIV by qPCR, “subclinical cases” were defined as those horses not exhibiting clinical signs of respiratory disease while testing positive for EIV by qPCR, and “control horses” did not show clinical signs of respiratory disease and were negative for EIV by qPCR.
Line 65 How was material recovered from the nasal swabs for testing?
Line 67 How was material recovered from the nose swipes for testing?
Line 70-76 How were the fluids recovered from the sponges? Was the total volume extracted for testing or a proportion of it?
Line 120 I would strongly suggest the authors provide the primary data for their study as a supplemental table that shows the qPCR result (cycle threshold) for each sample on every sampling day across the study. The supplemental table should also include the days the clinical cases exhibited signs of respiratory disease. Table 1 simply summarises these results and does not allow an assessment to be made of the dynamics of the outbreak. Furthermore, Table 1 would not permit future meta-analyses of the data or comparison of the data to that of future studies.
Line 149 suggest revision “6 inch”
I would also suggest that the Imperial measurements be quoted in metric measurements. However, I suspect these are as provided by the manufacturer and therefore their use is acceptable, given it is a relatively simple process to convert.
Author Response
The authors thank the reviewer for the very valuable comments and suggestions. All the issues have been carefully addressed, and the authors hope that they have succeeded in addressing the comments and suggestions.
Line 16 suggest revision “EI virus (EIV)”
The sentence was changed as suggested by the reviewer.
Line 55 suggest adding the day the suspected index case first tested positive by qPCR.
The date of November 14th was added to the manuscript.
Line 62 I would suggest the authors add clear definitions for each of the three groups of horses. My understanding is “cases” were defined as those horses exhibiting clinical signs of respiratory disease and testing positive for EIV by qPCR, “subclinical cases” were defined as those horses not exhibiting clinical signs of respiratory disease while testing positive for EIV by qPCR, and “control horses” did not show clinical signs of respiratory disease and were negative for EIV by qPCR.
The reviewer is correct in defining the case definition. Case definition is defined in material and methods as follows: Based on the EIV qPCR results and presence or absence of clinical signs, the 27 horses were placed in one of three groups: clinical EI (n=5), subclinical EI (6) and healthy group (16). Additional information has been provided to the manuscript to avoid any confusion.
Line 65 How was material recovered from the nasal swabs for testing?
Nasal swabs were collected according to standard collection methods, by advancing the 6” rayon-tipped swab into the ventral nasal meatus to collect rostral nasal secretions. Once collected, the swabs were placed in a sterile tube and kept refrigerated until processed. Additional information has been added to the manuscript.
Line 67 How was material recovered from the nose swipes for testing?
The nose wipes were vortexed and squeezed so to recover the saline solution containing the nasal secretions. Additional information has been added to the manuscript.
Line 70-76 How were the fluids recovered from the sponges? Was the total volume extracted for testing or a proportion of it?
The sponges in the bag were massages to squeeze the fluid out of the sponge and small aliquot was recovered for procession. The missing information has been added to the manuscript.
Line 120 I would strongly suggest the authors provide the primary data for their study as a supplemental table that shows the qPCR result (cycle threshold) for each sample on every sampling day across the study. The supplemental table should also include the days the clinical cases exhibited signs of respiratory disease. Table 1 simply summarises these results and does not allow an assessment to be made of the dynamics of the outbreak. Furthermore, Table 1 would not permit future meta-analyses of the data or comparison of the data to that of future studies.
The authors strongly agree with the reviewer that table 1 only summarizes the data. In order to bring transparency to the data, the authors have added a second table showing the overall agreement and discrepancy between the various sample types.
Line 149 suggest revision “6 inch”
The missing information has been added to the manuscript.
I would also suggest that the Imperial measurements be quoted in metric measurements. However, I suspect these are as provided by the manufacturer and therefore their use is acceptable, given it is a relatively simple process to convert.
The conversion in centimeters has been added for the 6-inch swab.
Reviewer 2 Report
Comments and Suggestions for Authors
General comments
This case study describes the comparison of different sampling types for qPCR analysis in an outbreak of equine influenza at a veterinary hospital. The information is highly relevant and interesting.
The reviewer’s main criticism is the missing definition of “agreement” between samples and lacking description of how this was calculated. Presumably, all sample pairs were viewed and the percentage of “agreeing” samples (both negative or both positive) was determined as part of the total; however, this is not described. In addition, the “disagreement” between samples (one positive, one negative) is not described further, such that the reader does not know whether nasal swabs were positive when other samples were not, or whether there were instances where nasal swabs were negative while other samples were positive. The authors are asked to include the method of calculation of agreement and it is strongly suggested that additional information on the sample pairs without agreement is included. Even without addressing the different time points, a simple table could be included for each the clinically and subclinically affected horses, indicating the number of samples in each cell:
Nasal swab |
Nose wipe |
Stall sponge |
Air sample |
|||
|
+ |
- |
+ |
- |
+ |
- |
+ |
|
|
|
|
|
|
- |
|
|
|
|
|
|
Alternatively, the information on the number of samples that were both positive, both negative, positive/negative or negative/positive should be included in the main text.
Specific comments
Line 33: Please change to “morbidity rate for EIV infection” or “morbidity rate for EI”.
Line 43/44: Please change to “for the testing for equine herpesviruses”.
Line 53: Please change to “The EI suspected index case”.
Line 70: Please clarify which “samples” this refers to – the nose wipes or the nasal swabs or all samples? (“Samples were then placed into…..”). If this refers to all samples, it might be best to place this sentence at the end of the paragraph.
Line 96-98: Please indicate when, in reference to the outbreak, this horse was admitted – was the horse on the premises prior to detection of the index case or added after EIV infection had been diagnosed?
Line 116 and 134: Please clarify how “agreement” was calculated. It would also be interesting to know if “disagreement” always meant that nasal swabs were positive while the other sample type was negative, or if instances occurred where it was the other way around. See general comments.
Author Response
The authors thank the reviewer for the very valuable comments and suggestions. All the issues have been carefully addressed, and the authors hope that they have succeeded in addressing the comments and suggestions.
The reviewer’s main criticism is the missing definition of “agreement” between samples and lacking description of how this was calculated. Presumably, all sample pairs were viewed and the percentage of “agreeing” samples (both negative or both positive) was determined as part of the total; however, this is not described. In addition, the “disagreement” between samples (one positive, one negative) is not described further, such that the reader does not know whether nasal swabs were positive when other samples were not, or whether there were instances where nasal swabs were negative while other samples were positive. The authors are asked to include the method of calculation of agreement and it is strongly suggested that additional information on the sample pairs without agreement is included. Even without addressing the different time points, a simple table could be included for each the clinically and subclinically affected horses, indicating the number of samples in each cell.
The method of calculation for agreement has been added and consisted in agreement (positive and negative) of paired samples.
The authors have also added Table 2 showing the number of samples that agreed and also disagreed with each other.
Specific comments
Line 33: Please change to “morbidity rate for EIV infection” or “morbidity rate for EI”.
The sentence has been changed.
Line 43/44: Please change to “for the testing for equine herpesviruses”.
The changes have been included in the manuscript.
Line 53: Please change to “The EI suspected index case”.
The sentence was changed as suggested by the reviewer.
Line 70: Please clarify which “samples” this refers to – the nose wipes or the nasal swabs or all samples? (“Samples were then placed into…..”). If this refers to all samples, it might be best to place this sentence at the end of the paragraph.
More information was added to avoid any confusion and specifically address the sample type mentioned in the text.
Line 96-98: Please indicate when, in reference to the outbreak, this horse was admitted – was the horse on the premises prior to detection of the index case or added after EIV infection had been diagnosed?
This case was unrelated to the hospital outbreak and presented to the VMTH about one week after the onset of the hospital outbreak. More information was added in the manuscript.
Line 116 and 134: Please clarify how “agreement” was calculated. It would also be interesting to know if “disagreement” always meant that nasal swabs were positive while the other sample type was negative, or if instances occurred where it was the other way around. See general comments.
Additional information was added to determine how the agreement was calculated. A table was added as well. While most of the disagreement was between a positive nasal swab and other negative sample types in horses from the clinical EI group, the disagreement went both directions for the subclinical EI group. Additional information has been added in the manuscript.